# Influence of Fly Ash on the Fluidity of Blast Furnace Slag for the Preparation of Slag Wool

Peipei Du [1], Yue Long [2,*], Yuzhu Zhang [1,2] and Liangjin Zhang [2]

1   School of Metallurgy, Northeastern University, Shenyang 110819, China
2   School of Metallurgy and Energy, North China University of Science and Technology, Tangshan 063009, China
*   Correspondence: longyue@ncst.edu.cn; Tel.: +86-15081952158

**Abstract:** Using fly ash as the modifier, blast furnace slag was modified to prepare slag wool, fulfilling the goal of using one type of waste to make use of another type of waste, and it is of great significance for the comprehensive utilization of industrial bulk solid wastes and resource recycling. In the process of forming fiber from blast furnace slag, fluidity is the key factor affecting the smooth formation of fiber from slag. To explore the changes in the fluidity of modified blast furnace slag, the temperature-dependent viscosity of modified blast furnace slag with different amounts of fly ash added was measured, and the effects of fly ash addition on the viscosity, fluidity, and activation energy of particle migration, and slag structure of modified blast furnace slag were investigated. The results indicated that with the increase in the amount of fly ash added, in the high-temperature region (>1324 °C), the viscosity of modified blast furnace slag increases gradually, the fluidity decreases gradually (i.e., the fluidity becomes worse), and the suitable fiber-forming temperature range gradually widens. When the fly ash addition increases from 5% to 25%, the trend of the activation energy of slag particle migration is as follows: increase, decrease, increase significantly, decrease. When the addition of fly ash is less than 20%, the $SiO_2$ content and slag temperature jointly affect the breakage and reorganization of oxygen bridge bonding in the silicon-oxygen tetrahedron in the slag structure. When the addition of fly ash increases to 25%, the slag temperature dominates the breakage of oxygen bridge bonding in the silicon-oxygen tetrahedron in the slag structure. When using fly ash as the modifier to prepare slag wool, from the perspective of slag fluidity and process operability, the optimum addition amount of fly ash is 15%.

**Keywords:** fly ash; blast furnace slag; fluidity; activation energy; viscosity





## 1. Introduction

Molten silicates discharged from blast furnaces have a chemical composition range of (36~42) wt.% $SiO_2$, (38~49) wt.% CaO, (1~13) wt.% MgO and (6~17) wt.% $Al_2O_3$, and a small amount of sulfide [1]; blast furnace slag can be used as a high-quality cement or directly replace part of the cement for concrete production after water quenching and cooling and has gradually become a high-quality primary resource [2], with a 100% utilization rate [3]. In recent years, the utilization of blast furnace slag has been more inclined toward products with high economic value, such as glass ceramics and refractory materials [4,5], but the problem of sensible heat utilization and recovery of molten blast furnace slag has not been solved. Slag wool is prepared using silicate industrial waste as the main raw material with the characteristics of light weight; they have a specific chemical composition range: 36~42 wt.% $SiO_2$, 28~47 wt.% CaO, 3~12 wt.% MgO, and 9~17 wt.%$Al_2O_3$ [6], and low thermal conductivity, corrosion resistance, good chemical stability, good sound absorption performance, etc. Slag wool can be used as an insulating filling material for buildings [7], sound absorption [8] and sound insulation material and insulating filling material for various equipment fillings [9]. Therefore, the blast furnace slag can be reconstituted to prepare slag wool. Using modified blast furnace slag to produce slag wool can realize not

only the high value-added utilization of blast furnace slag but also the effective use of the sensible heat of slag [10,11], which is a practical and effective way to realize the scale utilization of high value-added products made of blast furnace slag [12].

Fly ash is the main solid waste discharged from coal-fired power plants and one of the industrial waste residues with a large discharge capacity. Its main oxide composition is $SiO_2$, $Al_2O_3$, $FeO$, $Fe_2O_3$, $CaO$, $TiO_2$, etc [13]. At present, fly ash is mostly used for mine backfilling, road building, reclamation, building materials, concrete and other aspects [14]. From the chemical composition, fly ash is used to prepare slag wool by conditioning blast furnace slag, which provides a practical and effective way for the utilization of fly ash, thus fulfilling the goal of using one type of waste to make use of another type of waste, and it is of great significance for comprehensive utilization of industrial bulk solid wastes and resource recycling. At present, the mainstream method for producing slag wool is the centrifugal method. After modification, the blast furnace slag is dropped on the centrifugal roller of a four-roll centrifuge using intermediate equipment and is rapidly drawn and solidified to form fibers by high-speed rotation in the centrifuge [15,16]. This process has high requirements on the properties of modified blast furnace slag, including crystallization performance and fluidity; fluidity is the key factor affecting whether the slag can be successfully made into fibers [17,18]. At present, the influence of fly ash on the properties of blast furnace slag during the preparation of slag wool has been partially studied. Ren [19] used fly ash to modify blast furnace slag and studied the performance changes during the modified process. The addition of fly ash could optimize the high-temperature viscosity and crystallization performance of blast furnace slag. Additionally, the crystallization temperature of the slag decreased with increasing fly ash addition, and the suitable fiber-forming temperature range first increased and then decreased. Tang [20] used blast furnace slag and fly ash as raw materials, based on a $CaO$-$SiO_2$-$Al_2O_3$-$MgO$-$FeO$-$Na_2O$-$K_2O$ multi-component system, to investigate the melting behavior of quenched and tempered slag at different blast furnace slag ratios. The study found that with the reduction in the blast furnace slag ratio, liquidus temperature and flow point temperature first decreased and then increased. Yao [21] studied the crystallization behavior of quenched and tempered blast furnace slag in the cooling process with different fly ash content. The theoretical crystallization temperature of quenched and tempered blast furnace slag decreased gradually with the increase in fly ash content; When the addition of fly ash is higher than 15%, the quenched and tempered high slag is still glass phase when the temperature drops to 1000 °C and crystallization does not occur. At this time, slag crystallization is no longer the limiting link of slag wool molding. Most of these studies focused on the influence of fly ash on the crystallization performance of blast furnace slag, and few systematic reports discussed the influence of modified blast furnace slag on its fluidity. In this study, fly ash was selected as the modifier, and the mechanism of the fluidity changes in the modified blast furnace slag was analyzed in depth in terms of the slag viscosity, fluidity, activation energy of particle migration and structure; the influence of fly ash on the blast furnace slag fluidity was clarified, aiming to provide theoretical guidance for the preparation of high-quality slag wool from blast furnace slag.

## 2. Materials and Methods

### 2.1. Raw Materials

The raw materials were dry blast furnace slag and fly ash. After natural cooling of the blast furnace slag, dry blast furnace slag was obtained and crushed to <3 mm for later use. The fly ash used in the experiment is from the power plant, and it meets the index requirements of GB 18599—2020 <Standard for Pollution Control on Storage and Landfill of General Industrial Solid Wastes>, HJ 761—2015 <Determination of Organic Matter in Solid Wastes—Ignition Loss Method>, and NY/T 1121.16—2006 <Soil Testing—Determination of Total Water Soluble Salts in Soil> [22–24]. The main chemical components of the dry blast furnace slag and fly ash are listed in Table 1.

**Table 1.** Main chemical components of the raw materials (wt.%).

| Raw Material | SiO$_2$ | CaO | MgO | Al$_2$O$_3$ | K$_2$O | Na$_2$O |
|---|---|---|---|---|---|---|
| Blast furnace slag | 32.60 | 36.43 | 8.72 | 15.44 | 0.71 | 0.55 |
| Fly ash | 51.87 | 3.37 | 0.011 | 33.34 | 0.78 | 0.10 |

Table 1 shows that the content of SiO$_2$ in the fly ash is relatively high, while the contents of CaO and MgO are relatively low. According to the requirements of high Si and moderate Ca contents in the raw material composition of slag wool [6], the fly ash was used to modify the blast furnace slag, which can effectively reconstruct the composition of blast furnace slag as the raw material for slag wool, thus fulfilling the goal of using one type of waste to make use of another type of waste.

The amount of fly ash added was used as the experimental variable. The amount of fly ash added during the experiment was 5%, 10%, 15%, 20%, and 25%. The main chemical components of different modified blast furnace slags were calculated as listed in Table 2, where R$_2$O is the total content of the alkali metal oxides Na$_2$O and K$_2$O in the slag.

**Table 2.** Main chemical components of the raw materials (wt.%).

| Sample | Chemical Composition of Modified Blast Furnace Slag | | | | |
|---|---|---|---|---|---|
| | SiO$_2$ | CaO | MgO | Al$_2$O$_3$ | R$_2$O |
| Modified blast furnace slag 1 | 33.56 | 34.78 | 8.28 | 16.34 | 1.24 |
| Modified blast furnace slag 2 | 34.53 | 33.12 | 7.85 | 17.23 | 1.23 |
| Modified blast furnace slag 3 | 35.49 | 31.47 | 7.41 | 18.13 | 1.20 |
| Modified blast furnace slag 4 | 36.45 | 29.82 | 6.98 | 19.02 | 1.17 |
| Modified blast furnace slag 5 | 37.42 | 28.17 | 6.54 | 19.92 | 1.18 |

*2.2. Viscosity Test*

During the formation of slag wool, the modified blast furnace slag is a Newtonian fluid in a high-temperature state, and the fluidity can be characterized by the melt viscosity [25]. The viscosity of the modified blast furnace slag was measured by an RTW-08-type physical characteristic identifier, and the parameters of the instrument were calibrated with silicone oil at room temperature. A total of 140 g of sample was weighed and placed into a graphite crucible. The crucible containing the sample was placed inside the tester and heated to 1500 °C, and then, the temperature was held for 30 min. Subsequently, the viscosity was measured at different temperatures with a temperature drop step of 3 °C/min. The schematic diagram of the experimental setup is shown in Figure 1.

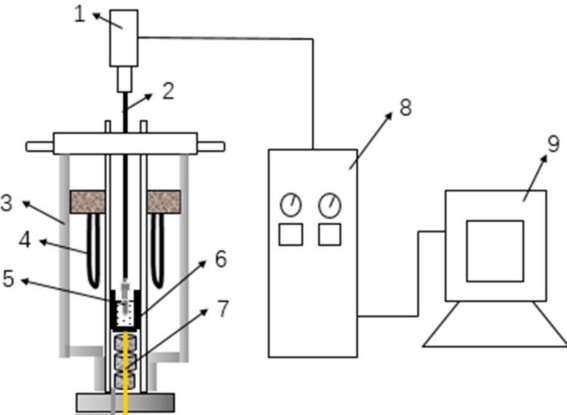

**Figure 1.** Schematic diagram of the viscosity test. 1-torque sensor 2-corundum rod 3-furnace 4-silicon molybdenum heating element 5-molybdenum rotor 6-graphite crucible 7-thermocouple 8-control panel 9-computer.

### 2.3. Viscosity Model

The Vogel–Fulcher–Tammann (VFT) model in Equation (1) was adopted as the viscosity-temperature model of the high-temperature region of the modified blast furnace slag [26]. where $\eta$ is the viscosity of the modified blast furnace slag when the temperature is $T$, Pa·s; A and B are constants related to the composition of the modified blast furnace slag and $T_0$ is a temperature constant. All parameters can be obtained by fitting the experimental data, and the VFT model is accurate for fitting the high-temperature region.

$$\log \eta = A + \frac{B}{T - T_0} \tag{1}$$

### 2.4. Activation Energy of Particle Migration

In the high-temperature region where the modified blast furnace slag is suitable for fiber formation, the movement of each particle in the slag is controlled by its adjacent particles. Only when the particle has enough energy ($\Delta E$) to overcome the attraction of the surrounding particles can it migrate effectively, showing microfluidity. The more activated the particles are, the greater the fluidity. The relationship between the viscosity and temperature of the modified blast furnace slag is characterized by the Arrhenius equation [27] where $\Delta E$ is the particle viscous activation energy in the melt, J/mol; $\eta_0$ is a constant related to the composition of the melt; $k$ is the Boltzmann constant, J/K and $T$ is the absolute temperature, K. Taking the logarithm of both sides of Equation (2) gives. Equation (3) indicates that $\eta$ is linearly related to $\frac{1}{T}$, and the slope is $\frac{\Delta E}{k}$.

$$\eta = \eta_0 \exp(\frac{\Delta E}{kT}) \tag{2}$$

$$\ln \eta = \ln \eta_0 + \frac{\Delta E}{k} \frac{1}{T} \tag{3}$$

## 3. Results and Discussion

### 3.1. Influence of Fly Ash on the Viscosity and Fluidity of Modified Blast Furnace Slag

In the fiber-forming process of blast furnace slag, the melt is subject to complex external forces, the relative motion between the liquid layers can occur, and the internal friction against this movement is the melt viscosity. Therefore, the melt viscosity can indirectly reflect the fluidity, and melts with a high viscosity do not easily flow. Figure 2a shows the viscosity-temperature curves of modified blast furnace slag with different fly ash additions. The viscosity of blast furnace slag with different fly ash additions increases with decreasing temperature since the decrease in temperature reduces the number of particles with viscous activation energy in the slag, and the weakening of thermal vibrations leads to complex ion aggregation. When the amount of fly ash added is 5%, an inflection point appears in the viscosity-temperature curve of the blast furnace slag at approximately 1338 °C, and when the temperature is 1324 °C, the viscosity of the slag increases sharply (almost a vertical line in the figure), showing the characteristics of short slag. When the addition amount of fly ash is 10–25%, the viscosity-temperature curves of the blast furnace slag are smooth without obvious inflection points, and the viscosity change rate is small, showing the characteristics of long slag. In addition, in the high-temperature region (temperature > 1324 °C), for a single temperature, the viscosity of the slag gradually increases with the increase in the amount of fly ash added. According to the ionic structure theory of silicate melts [28], the size of the Si-O anion group, i.e., the network former with the highest content in the blast furnace slag, is considerably larger than that of the cation, making the Si-O anion group the main viscous flow unit in the slag. In the high-temperature region (temperature > 1324 °C), the addition of fly ash increases the $SiO_2$ content in the slag, which increases the possibility of Si-O ion agglomeration. Therefore, the viscosity of the slag gradually increases with increasing fly ash content. In addition, the fluidity of the slag also gradually deteriorates due to the aggregation of siloxane groups.

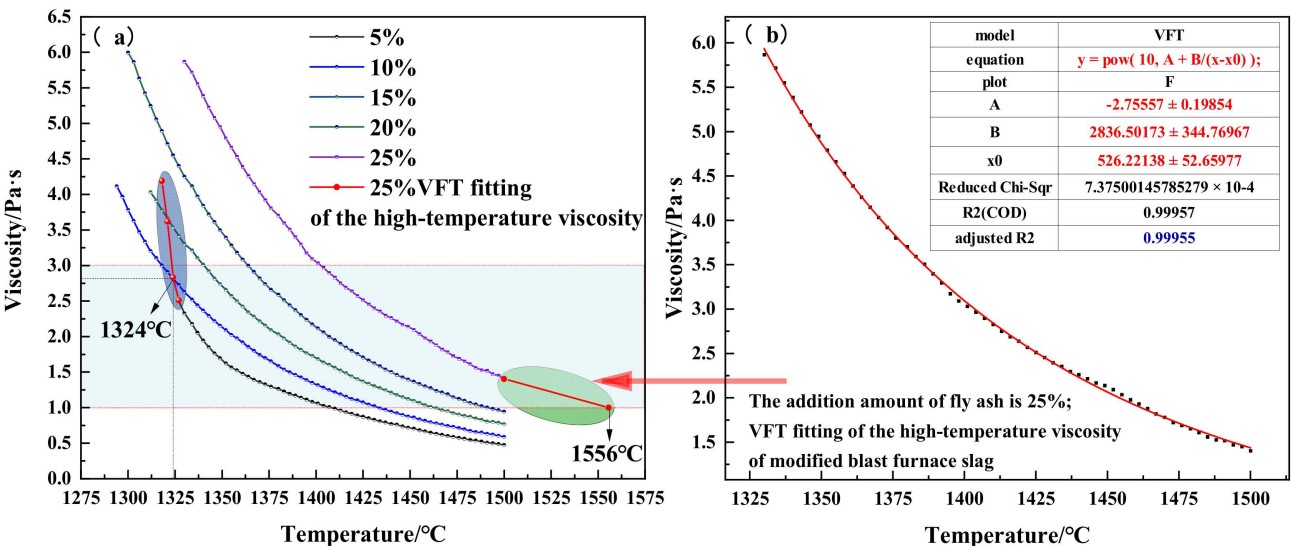

**Figure 2.** (**a**) Viscosity-temperature curve of modified blast furnace slag (**b**) and its VFT fitting when the amount of fly ash added is 25%.

The reciprocal of the viscosity $\varphi = 1/\eta$ is used to characterize the difficulty of slag flow and is called the fluidity of slag. The greater the fluidity is, the better the slag fluidity. Figure 3 shows the curve of the fluidity of modified blast furnace slag with different fly ash additions as a function of temperature. Figure 3 indicates that in the high-temperature region (temperature > 1324 °C), when the addition amount of fly ash increases from 5% to 25%, the fluidity of the modified blast furnace slag increases with increasing temperature, i.e., the fluidity improves. At a single temperature, the fluidity of the slag decreases with the increase in the amount of fly ash added, i.e., the fluidity of the slag worsens. With increasing temperature, the difference in the fluidity of the slag tends to increase. When the temperature is <1324 °C, the fluidity of the modified blast furnace slag with 5% fly ash addition tends to decrease sharply and is less than the fluidity of the modified blast furnace slag with 10% fly ash addition. When the temperature is <1324 °C, the liquid-solid transformation of the modified blast furnace slag with 5% fly ash addition is violent and many solids appear in the slag, which makes the fluidity deteriorate significantly. In the blast furnace slag, silicon oxide and aluminum oxide ions are the main viscous flow units. The addition of fly ash increases the $SiO_2$ content in the slag, which increases the possibility of Si-O ion aggregation. Therefore, as shown in Figure 3, the fluidity of the slag gradually decreases with increasing fly ash content, and the fluidity of the slag gradually deteriorates due to Si-O ion aggregation. Similarly, with increasing temperature, the voids of the slag increase, which is conducive to the interpenetration and movement of silicon oxide ion groups; thus, the fluidity of the slag increases, i.e., the fluidity improves.

### 3.2. Influence of Fly Ash on the Activation Energy of Particle Migration of Modified Blast Furnace Slag

Generally, the melt viscosity of the raw materials required to produce slag wool should be between 1 and 3 Pa·s [6], and at the same time, good fluidity should be guaranteed so that the raw materials can be in a free-flowing state, that is, the slag temperature should be larger than its melting temperature ($T_m$) [29]. Figure 2a demonstrates that when the addition amount of fly ash increases to 25%, the viscosity of blast furnace slag is 1.403 Pa·s (>1 Pa·s) at 1500 °C. With increasing temperature, the viscosity-temperature relationship satisfies the VFT equation. The VFT fitting of the viscosity-temperature curve when the fly ash addition amount is 25% is shown in Figure 2b; the goodness of fit $r^2$ is 0.99957. According to the fitting curve in Figure 2b, when the amount of fly ash is 25%, the temperature of modified blast furnace slag at a viscosity of 1 Pa·s is 1556 °C. Since the melting temperature is defined as the temperature at the intersection of the viscosity-temperature curve and the

45° tangent line, the melting temperature of the modified blast furnace slag can be obtained through the viscosity-temperature curve. The temperature and melting temperature of modified blast furnace slag with different fly ash additions at viscosities of 1 Pa·s and 3 Pa·s are shown in Figure 4a; the suitable fiber-forming temperature range is shown in Figure 4b. With the addition amount of fly ash being from 5% to 25%, the suitable temperature range for fiber formation gradually widens from 62 °C to 152 °C, a significant increase.

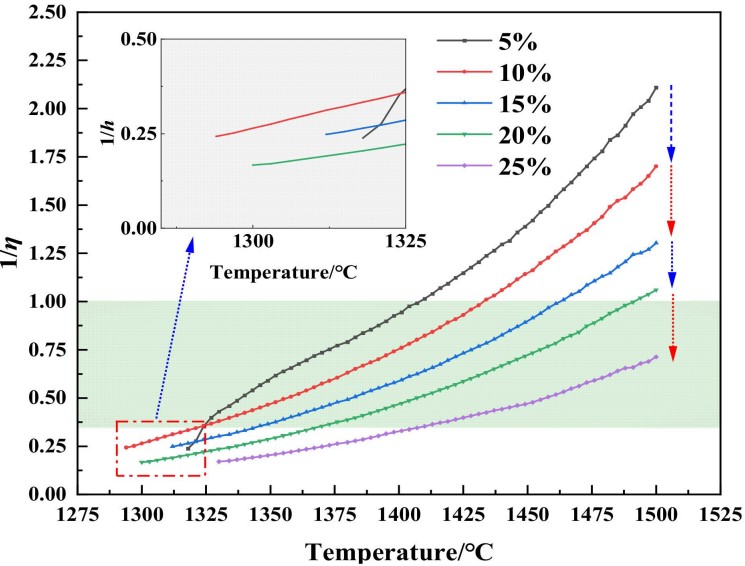

**Figure 3.** Curve of the fluidity of modified blast furnace slag as a function of temperature.

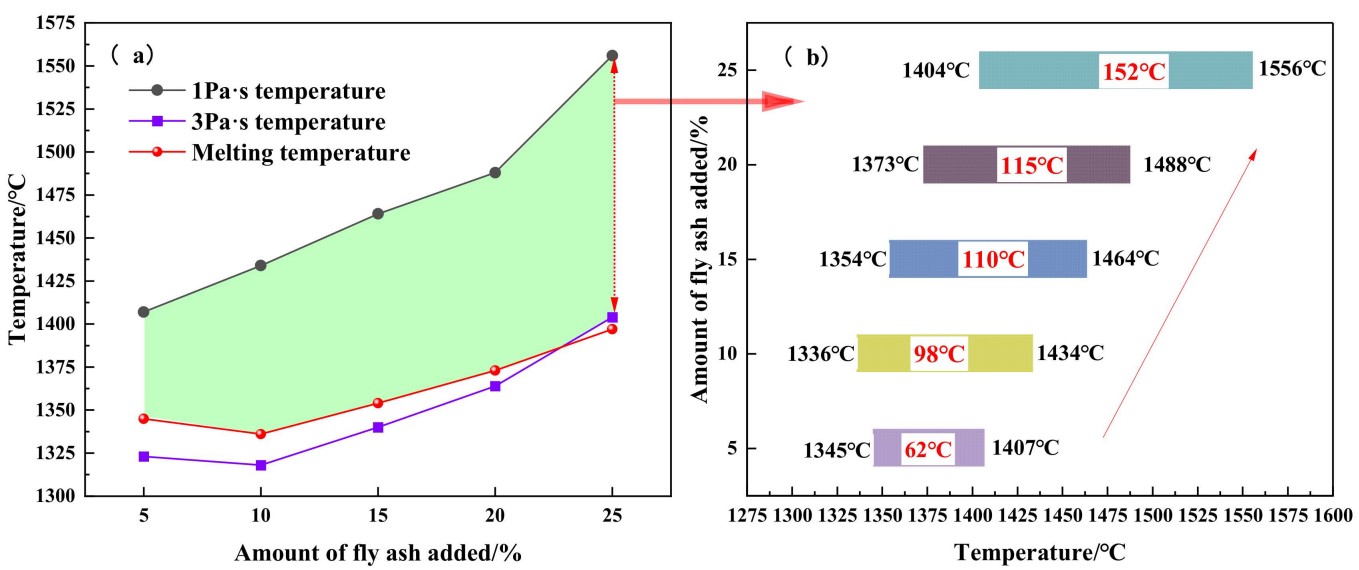

**Figure 4.** Changes in the suitable fiber-forming temperature range of modified blast furnace slag. (**a**) The temperature and melting temperature of modified blast furnace slag with different fly ash additions. (**b**) The suitable fiber-forming temperature range.

According to the suitable fiber-forming temperature range of modified blast furnace slag with different amounts of fly ash added in Figure 4b, the relationship between $\ln \eta$ and $\frac{1}{T}$ was fitted, as shown in Figure 5, and $r^2$ is the goodness of fit. Figure 5 demonstrates that under different fly ash additions, there is an obvious linear relationship between $\ln \eta$ and $\frac{1}{T}$ for modified blast furnace slag, and the correlations are all greater than 0.99. According to the slope of the fitted straight line, the activation energy of particle migration of the modified blast furnace slag was calculated, as shown in Figure 6a.

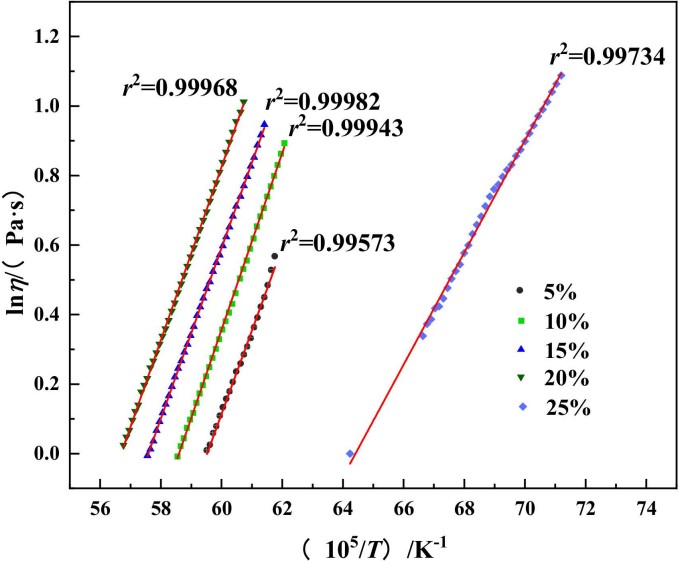

**Figure 5.** Fitting curve of $\ln \eta$ and $\frac{1}{T}$ of modified blast furnace slag.

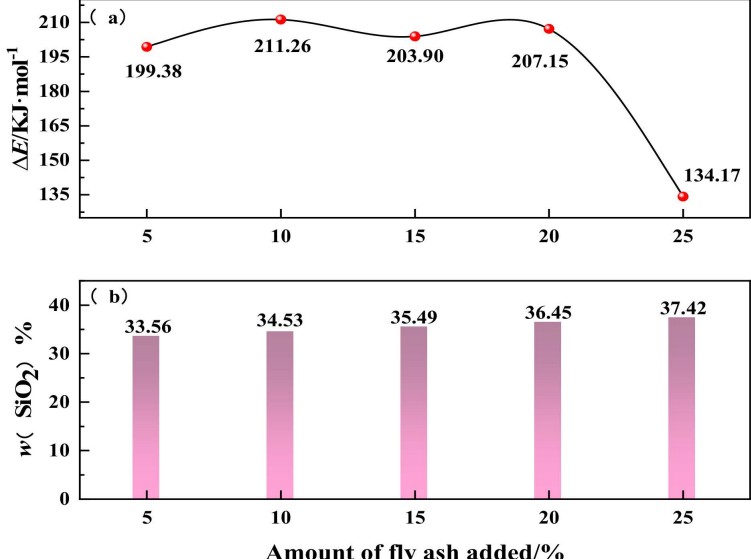

**Figure 6.** (**a**) Activation energy of particle migration of modified blast furnace slag (**b**) and the SiO$_2$ content with the addition of fly ash.

Figure 6a indicates that when the addition amount of fly ash is less than 20%, the activation energy of particle migration in the modified blast furnace slag fluctuates around 200 kJ/mol with small changes, increasing initially and then decreasing overall. When the addition amount of fly ash increases to 25%, the activation energy of particle migration of the slag is greatly reduced to 134.17 kJ/mol. According to the theory of the silicate melt structure [30], the activation energy of particle migration of modified blast furnace slag is related to not only the slag composition but also the degree of aggregation of the aggregates in the slag, which consists of numerous silicone complex anion groups of different sizes. The variation in the SiO$_2$ content of modified blast furnace slag with fly ash addition in Figure 6b demonstrates that with the increase in fly ash addition, the SiO$_2$ content in the modified slag increases, and the increased amount of Si$^{4+}$ leads to an increase in the large aggregate concentration consisting of silicon-oxygen and aluminum-oxygen complex anion groups, which is also the main reason for the increase in the activation energy of slag particle migration when the fly ash addition increases from 5% to 10%. When the addition amount of fly ash increases to 15%, the content of SiO$_2$ in the slag further increases, but the

suitable fiber-forming temperature of the modified slag is high, and the high temperature leads to an increase in the concentration of small aggregates consisting of silicone complex anion groups. Among the limiting factors of the aggregate type in the slag, the influence of the temperature is larger than the composition, resulting in a small reduction in the activation energy of slag particle migration. When the addition amount of fly ash increases to 20%, the content of $SiO_2$ in the slag further increases to 36.45%. The increase in the large aggregate concentration in the slag caused by the increase in $Si^{4+}$ is greater than that of the small aggregate concentration caused by the temperature, resulting in a small increase in the activation energy of slag particle migration. When the addition amount of fly ash increases to 25%, the optimum temperature for fiber formation of modified slag is 1404 °C, and the aggregate type in the slag is mainly affected by temperature. The concentration of small aggregates consisting of silicone complex anion groups in the slag increases significantly, leading to a significant reduction in the activation energy of particle migration, as shown in Figure 6a.

The lower the activation energy of particle migration is, the easier it is for the particles in the slag to overcome the attraction of the surrounding particles to migrate and flow. When fly ash is used as the modifier to prepare slag wool, the activation energy of slag particle migration is low when the addition amount of fly ash is 15%; the suitable fiber-forming temperature range is increased by 12.24% compared to that when the addition amount of fly ash is 10%. In actual production, when fly ash is used as the modifier to prepare slag wool considering the energy consumption and operability and the suitable fiber forming conditions should be determined with the consideration of the fluidity of the slag. Therefore, to better control the fluidity of the modified slag, the optimum addition amount of fly ash is 15%.

### 3.3. Effect of Fly Ash on the Slag Structure

The structure of modified blast furnace slag could directly affect the fluidity, and its basic structural unit is a silicon-oxygen tetrahedron. If alkali metals are introduced into the silicon-oxygen tetrahedral structure, since the R-O bond is weaker than the Si-O bond, $Si^{4+}$ could attract the oxygen ions on the R-O bond, which could cause the bridging oxygen in the silicon-oxygen tetrahedral structure to break, forming nonbridging oxygen, as shown in Figure 7.

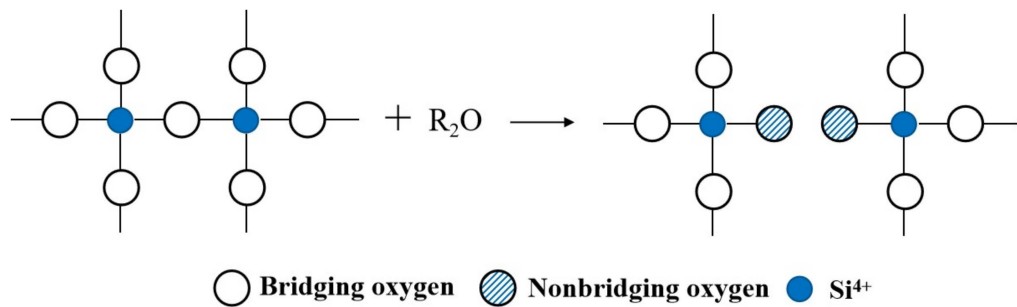

**Figure 7.** Schematic diagram of the breaking of oxygen bridge-bonding in the silicon-oxygen tetrahedron.

The breaking of oxygen bridge-bonding leads to a change in the bond length, bond angle and bond strength of the Si-O bond, thereby changing the connection mode, resulting in the formation of small aggregates partially formed by the short chains of the silicon-oxygen tetrahedra, and the generation of small aggregates could directly affect the activation energy of particle migration of the modified blast furnace slag, resulting in a change in the fluidity. This process is not stable, and the small aggregates could interact and merge to release oxides, as shown in Equation (4).

$$[SiO_4]R_4 + [Si_2O_7]R_6 = [Si_3O_{10}]R_8 + R_2O$$

$$2[Si_3O_{10}]R_8 = [SiO_3]_6R_{12} + 2R_2O \tag{4}$$

The released oxides could further erode the silicon-oxygen tetrahedron until the two processes reach equilibrium, and at this time, there are small aggregates, large aggregates, some lattice fragments, and some unreacted free oxides in the slag. According to the theory of the melt structure [28], the structural change in the silicon-oxygen tetrahedron in the blast furnace slag is related to the slag composition and temperature. When the addition amount of fly ash is between 5% and 10%, the increase in the $SiO_2$ content dominates the increase in the degree of aggregation of silica-oxygen tetrahedra in the slag. When the addition of fly ash is 15% in the slag, the oxygen bridge bonding in the silicon-oxygen tetrahedron is broken due to the high temperature. When the addition of fly ash increases to 20%, due to the high temperature and the increase in the $SiO_2$ content, the condensation phenomenon, i.e., the breakage and reorganization of oxygen bridge-bonding in the silicon-oxygen tetrahedron, occurs in the slag. When the fly ash addition increases to 25%, the temperature dominates the breakage of oxygen bridge bonding in the silicon-oxygen tetrahedron in the slag, showing the change in the activation energy of slag particle migration in Figure 6a.

## 4. Conclusions

Using fly ash as the modifier, blast furnace slag was modified to prepare slag wool, which is a practical and effective way to realize the scale utilization of high value-added products made of blast furnace slag and fly ash. However, fluidity is the key factor affecting the smooth formation of fiber from slag. The mechanism of the fluidity changes in the modified blast furnace slag was analyzed in depth in terms of the slag viscosity, fluidity, and the activation energy of particle migration and structure; the influence of fly ash on the blast furnace slag fluidity was clarified, provided theoretical guidance for the preparation of high-quality slag wool from blast furnace slag and fly ash.

(1) When the amount of fly ash added is in the range of 5% to 25%, the viscosity of modified blast furnace slag gradually decreases with increasing temperature. At a single temperature, with the increase in the addition amount, the viscosity of the modified blast furnace slag increases, and the fluidity decreases, i.e., the fluidity worsens. When the amount of fly ash added is less than 20%, the activation energy of particle migration of the modified blast furnace slag increases slightly first, then decreases and finally increases, but the overall change is not large, and the activation energy fluctuates around 200 kJ/mol. When the amount of fly ash added increases to 25%, the activation energy of slag particle migration is greatly reduced to 134.17 kJ/mol. When the addition of fly ash is 15%, the activation energy of particle migration of the modified blast furnace slag is 203.9 kJ/mol, and the suitable fiber-forming temperature range is 110 °C.

(2) When the addition amount of fly ash is lower than 10%, the increase in the $SiO_2$ content dominates, leading to an increase in the concentration of large aggregates in the slag and an increase in the activation energy of slag particle migration. When the fly ash addition amount is 15%, the slag temperature promotes the breakage of oxygen bridge bonding in the slag structure, and the activation energy of slag particle migration decreases slightly. When the addition amount of fly ash increases to 20%, the condensation phenomenon, i.e., the breakage and reorganization of oxygen bridge-bonding in the silicon-oxygen tetrahedron, occurs in the slag, and the activation energy of slag particle migration increases slightly. When the addition amount of fly ash increases to 25%, the slag temperature dominates and promotes the breakage of oxygen bridge bonding in the slag, resulting in a significant reduction in the activation energy of slag particle migration.

In actual production, with consideration of the energy consumption and operability of fiber forming process, we take into account the suitable fiber forming conditions and fluidity of molten modified blast furnace slag; when using fly ash as a modifier to prepare slag wool, 15% is the best addition amount in terms. This conclusion is universal and can provide theoretical guidance for the modification of blast furnace slag and the preparation of slag wool from blast furnace slag and fly ash.

**Author Contributions:** Data curation, L.Z.; Methodology, P.D.; Project administration, Y.L.; Software, P.D.; Writing—original draft, P.D.; Writing—review and editing, Y.Z. All authors have read and agreed to the published version of the manuscript.

**Funding:** This work was funded by the National Science Foundation of China (51874138).

**Data Availability Statement:** Data sharing not applicable.

**Conflicts of Interest:** The authors declare no conflict of interest.

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
