# Peer review of "Influence of Fly Ash on the Fluidity of Blast Furnace Slag for the Preparation of Slag Wool"

_crystals, doi:10.3390/cryst13010119_

Round 1

Reviewer 1 Report

Paper is clearly written. Precise, and easy to understand. Results and analysis are adequate. Conclusions are clear. References are sufficient. 

Perhaps adding some content on sustainability or SDG on fly ash reutilization would be relevant

Author Response

Dear Reviewer and Editor,

We provided a point-by-point response to the reviewer’s comments .

Reviewer 2 Report

The work is quite interesting and considers a promising area of building materials and products. The relevance of the work is beyond doubt, due to the development of the building materials market and environmental issues.

However, I have some comments.

1 The list of references requires adjustments: I consider it unacceptable to make references to dissertations for various reasons, primarily due to the limitations of their search; one gets the impression that, apart from researchers from the People's Republic of China, no one was involved in the processing of ash, blast-furnace slag and the production of building materials from them. I ask the authors to also consider other works.

2. The authors pay great attention to the rheology of the material at certain temperatures. However, they give a very great influence to the material composition and consider only in this vein. In this connection, the authors are recommended to consider in more detail the initial products, especially ash: it is recommended to study their structure and granulometric characteristics. These studies will help to connect the parts into a whole. Also an interesting point is the structure of the material obtained, how high the quality of the material turned out.

3 The authors say nothing about natural radionuclides, and in general the radiation safety of the raw materials used. Ashes from coal combustion quite often have high showed.

4. Another note is that the authors talk about some interaction between slag and ash, but do not give sufficient experimental confirmation, such as fixing phase transitions.

In general, the article is highly commendable and, with additional corrections, may be published in Crystals magazine.

Author Response

(The authors gave the same response as above.)

Reviewer 3 Report

Paper ID: crystals-2121515

Type: Article 
Title: 
Influence of fly ash on the fluidity of blast furnace slag for the preparation of slag wool

Authors: Pei-pei Du , Yue Long , Yu-zhu Zhang , Liang-jin Zhang

 This study investigates fly ash on the fluidity of blast furnace slag to prepare slag wool. Although the testing methods and compared results attained in the present study show the importance of the paper, The authors should address the following comments: 

 Novelty in comparison to recent literature? Need to be emphasized.

2.     please highlight the novelty of the article in the abstract and the introduction,

  1. I suggest more improve the introduction section.
  2. please eliminate citation pockets and cite each reference individually, showing the scientific reason why each reference has been cited. If it is not possible to do it in this way, please remove unnecessary reference
  3. There should be a space between number and unit. Please correct these errors in the paper.
  4. Throughout the text, some typos must be eliminated.

Author Response

(The authors gave the same response as above.)

Reviewer 4 Report

The conducted work “Influence of fly ash on the fluidity of blast furnace slag for the preparation of slag wool” is good. However, following comments should be addressed to further improve the paper:

1.      Explicitly mention the novelty and research significance of current work in last paragraph of introduction section with emphasis on scientific soundness. Also, add recent relevant literature review more from 2022 papers in introduction section as there is only one paper cited from 2022.

2.      Avoid paragraph of few (1-4) sentences throughout the manuscript, particularly in results and discussions sections e.g. lines 187-189, 192-196, etc.

3.      Avoid long sentences throughout the manuscript, e.g. lines 240-244, etc.

4.      Results are explained in a descriptive way, thus results in current form look like a project report. Results should be further elaborated with scientific reasoning.

5.      A separate brief section (explaining the relevance of this research for practical implementation) may be added before conclusion section.

6.      Conclusions should be reflection of obtained results with scientific soundness. Conclusions are little long; these should be to the point as obtained from results. Closing remarks should be added at the end of conclusion section keeping in mind all conclusive bullet points.

7.      English Language should be improved throughout the manuscript.

B. SPECIFIC COMMENTS FOR IMPROVING FOCUSSED RESEARCH

1.      Introduction should be strengthened.

2.      Lines 73 and 83, please check same heading number and heading caption.

Author Response

(The authors gave the same response as above.)
